# Transforming scholarly landscapes: The influence of large language models on academic fields beyond computer science

Aniket Pramanick[1]*, Yufang Hou[2], Saif M. Mohammad[3], Iryna Gurevych[1]

**1** Department of Computer Science, Technical University of Darmstadt, Darmstadt, Germany, **2** IT:U Interdisciplinary Transformation University Austria, **3** National Research Council Canada, Ottawa, Canada

* aniketpramanick26@gmail.com

## Abstract

Large Language Models (LLMs) have ushered in a transformative era in Natural Language Processing (NLP), reshaping research and extending NLP's influence to other fields of study. However, there is little to no work examining the degree to which LLMs influence other research fields. This work *empirically and systematically examines the influence and use of LLMs in fields beyond NLP.* We curate 106 LLMs and analyze ~148k papers citing LLMs to quantify their influence and reveal trends in their usage patterns. Our analysis reveals not only the increasing prevalence of LLMs in non-CS fields but also the disparities in their usage, with some fields utilizing them more frequently than others since 2018, notably Linguistics and Engineering together accounting for ~45% of LLM citations. Our findings further indicate that most of these fields predominantly employ task-agnostic LLMs, proficient in zero or few-shot learning without requiring further fine-tuning, to address their domain-specific problems. This study sheds light on the cross-disciplinary impact of NLP through LLMs, providing a better understanding of the opportunities and challenges.

## Introduction

Modern science heavily relies on citing past research to build upon past ideas, situate the proposed work, reject old hypotheses, etc., thereby facilitating the dissemination of good ideas [1]. Some ideas are limited in the scope of their influence (being cited narrowly within a specific subfield). At the same time, others may have broad applicability, influencing not only an entire field such as Computer Science (CS) but also many disciplines beyond. Since the number of published papers is too large for manual review [2,3], there is growing work in automatically and quantitatively tracking the influence of ideas within and across fields [4,5].

Arguably, one of the most transformative ideas over the past decade is that of LLMs [6]. Originally proposed within NLP, LLMs have revolutionized almost all research areas within NLP itself [7,8]. Moreover, their utilization is not confined to NLP; other fields are leveraging LLMs as well [9,10]. While it is well recognized

**Data availability statement:** All code, scripts for dataset download, and results files used in this study are publicly available at: https://github.com/UKPLab/plosone2025-llm-trends. The dataset used in this study is separately accessible at: https://github.com/allenai/s2orc?tab=readme-ov-file.

**Funding:** This work has been funded by the German Research Foundation (Deutsche Forschungsgemeinschaft, DFG) as part of the Research Training Group KRITIS (No. GRK 2222). This work has also been funded by the LOEWE Distinguished Chair "Ubiquitous Knowledge Processing", LOEWE initiative, Hesse, Germany (Grant Number: LOEWE/4a//519/05/00.002(0002)/81), and by the "Open-Access-Publikationskosten" programme of the German Research Foundation (Deutsche Forschungsgemeinschaft, DFG). The funders had no role in the study design, data collection and analysis, the decision to publish, or the preparation of the manuscript.

**Competing interests:** The authors have declared that no competing interests exist.

that these models are being adopted outside of NLP, the full extent and nature of their usage remains unclear. Analyzing the widespread adoption and utilization of LLMs across various fields provides essential insights for promoting responsible AI practices.

Although recent studies have begun to explore the influence of LLMs within CS [11,12], their impact in other disciplines beyond CS is still largely unknown. While it is evident that non-CS fields are paying attention to LLMs [13], precise details about which fields are leveraging them and the specific purposes of their use are still to be understood.

The modes of influence are multifaceted and intricate in nature, which makes it challenging to empirically determine the degree of influence. In this work, we focus on a specific dimension of influence: the scientific impact that one field has on another [14]. One notable marker of this inter-field influence is citation [15]. Therefore, we propose that the degree to which a source field cites the works of a target field can serve as a rough indicator of their influence [16]. Although citation patterns are subject to biases, meaningful aggregate-level insights can still be obtained [17, 18].

While no universally accepted definition of LLMs exists [7], in this work, we take LLMs as foundational models [8] built on the transformer architecture [19], pretrained on massive textual datasets with over 100M parameters. We carefully curated a dataset of 106 well-cited LLM papers up to February 2024 including BERT [20], T5 [21], GPT-3 [22], PaLM [23], ChatGPT [24], and LLaMA [25]. We then *quantitatively* investigate:

(A) *Which non-CS fields are impacted by LLMs? And to what degree?* (section Breadth of LLM adoption)

(B) *How do the usage patterns of LLMs evolve over time within these non-CS fields?* (section Evolving usage patterns of LLMs)

We complement the above discussions with an additional *qualitative* analysis exploring:

(C) *In what contexts are LLMs applied within these non-CS fields?* (section Applications of LLMs in non-CS fields)

In addressing the questions above, we utilize the Semantic Scholar data [26] to construct our dataset comprising ~148k papers from 22 fields outside of CS, published between 2018 and February 2024, that cite LLMs. This dataset includes structured full texts extracted from the PDFs of the papers as well as their metadata, including the fields of study, year of publication, venue of publication, and author information. Additionally, we include similar data from the papers that originally introduced these 106 LLMs, together with ~273k LLM citations from the aforementioned papers outside of CS.

While the quantitative analyses (**A** and **B**) provide insights into the adoption of LLMs across diverse fields, they do not capture the nuances of how LLMs are being utilized within these fields. To gain deeper insights for **C**, we delve beyond metadata, exploring the content of the papers citing LLMs. Our qualitative examination of papers' contents not only illuminates diverse applications of LLMs outside of CS but also guides towards potential improvements and research directions for better LLM usability in various non-CS fields. Finally, we approximately assess and discuss the extent to which these papers, especially in fields like Biology, Medicine, Psychology, Law, etc., discuss ethical concerns associated with LLMs, such as hallucinations [27] or non-reproducible outputs [28], using a smaller human-annotated dataset.

In summary, our findings suggest that 1) linguistics, engineering, and medicine are the top three fields citing LLMs, and fields more closely aligned with CS tend to more readily adopt LLM technology compared to other domains (section Breadth of LLM adoption); 2) BERT continues to be the preferred LLM among non-CS fields, even in terms of average yearly citations. Moreover, many of these fields frequently cite task-agnostic models like GPT-3 [22] or LLaMA [25], which excel in few-shot settings without needing additional fine-tuning (section Evolving usage patterns of LLMs); and 3) non-CS fields mostly use LLMs to solve their domain-specific problems (section Applications of LLMs in non-CS fields). Our study is the first attempt to empirically and systematically investigate the impact and usage of LLMs in fields outside CS.

The aim of our study is not to establish causal claims about the transformative impact of LLMs in non-CS fields, but rather to provide a systematic descriptive mapping of their diffusion. By presenting large-scale empirical evidence of how LLMs are cited and applied across 22 fields, we offer a baseline for future in-depth studies. Our intended audience includes NLP researchers interested in interdisciplinary exchange as well as domain specialists who may benefit from understanding how LLMs are entering their fields. We hope this work can raise awareness, identify opportunities for tailored model development, and highlight gaps in ethical awareness, thereby supporting more effective and responsible cross-disciplinary collaboration.

## Related work

### Scientific trends analysis

Early work by [29] served as a catalyst for exploring scientific trends in NLP. Research in scientific trends analysis primarily spans three dimensions, from citation patterns to metadata and content analysis. A vast amount of research has been done into the study of citation patterns and the amalgamation of topological measures within citation networks to assess research trends [30–32]. Complementary to this, [33] did a content analysis, employing rhetorical framing to elucidate trend patterns. [34] and [35] take a metadata-driven approach, employing prevalence correlation to examine the interplay between publication topics and research grants. [36] apply causal techniques to explore how tasks, methods, datasets, and metrics interact within the context of the evolution of NLP research. [37] explores dataset usage patterns across research communities and observes distinct trends in dataset creation within NLP communities, shedding light on the nuanced dynamics of research resource utilization. While the majority of the previous research in Computer Science and NLP examines research trends within Computer Science and NLP, respectively, in this work, we analyze both citations and content to investigate the influence of LLMs (CS and NLP technology) on fields outside of Computer Science.

### NLP scientometrics

In parallel, NLP scientometrics has witnessed significant advancements in recent years as researchers strive to understand the landscape of NLP research and its evolution [38,39]. Prior studies have examined the engagement of NLP research with other disciplines, shaping the research trajectory [40,41]. While various studies have observed a rising trend in interdisciplinary interactions occurring both across fields and within subfields [42,43], this growth primarily manifests in connections between closely related domains, with limited increases in associations between traditionally distant research

areas, such as materials sciences and clinical medicine [44]. We add to this line of work by investigating how research fields that were traditionally distant from NLP are now incorporating LLM techniques to address their challenges.

### LLM abilities assessment

LLMs are a category of transformer-based language models distinguished by their size, typically consisting of hundreds of millions or even more parameters. They are trained on vast and diverse data [45]. These models exhibit remarkable proficiency in understanding and generating natural language, enabling them to excel in complex language-related tasks [46,47]. However, LLMs come with challenges, including the generation of plausible yet factually incorrect text [48]. Moreover, they can be manipulated to produce harmful content, raising ethical concerns about their misuse [49,50]. Our study does not assess LLM capabilities or risks but focuses on how non-CS fields use LLMs, and we examine ethical concerns related to LLMs based on the content of the papers within these fields.

## Research methodology

Our study adopts a large-scale descriptive scientometric approach to explore how Large Language Models (LLMs) have diffused across research fields beyond Computer Science (CS). We focus on three overarching research questions.

(A) *Which non-CS fields are impacted by LLMs, and to what degree?*
(B) *How have usage patterns of LLMs evolved over time within these fields?*
(C) *In what contexts are LLMs applied within these fields, and to what extent are ethical concerns acknowledged?*

To answer these questions, we detail our process of data collection, field classification, and analysis procedures.

### Data collection

We began by curating a list of 106 influential LLMs up to February 2024, drawing on the Stanford Ecosystem Graphs [51]. This resource systematically tracks widely cited foundation models and their associated publications.

 **Choosing the LLMs.** The Computer Science community has yet to establish a universally accepted definition of LLMs. In this study, we operationalize LLMs as foundation models: transformer-based architectures pretrained on massive textual datasets, with more than 100M parameters. Proprietary LLMs without identifiable academic publication records were excluded from the bibliometric analysis, though they may still appear in qualitative analyses. This approach prioritizes models with sufficient coverage in bibliographic data to support reproducible large-scale analysis. We note that our curated set naturally emphasizes well-established, widely documented models such as BERT. In contrast, more recent instruction-tuned, multi-modal, or domain-specific LLMs were excluded because (a) they do not always have an identifiable associated publication, (b) their bibliometric coverage remains limited or inconsistent across databases, and (c) their short citation history prevents meaningful longitudinal trend analysis.
Using Semantic Scholar Dataset (S2) [52], we retrieved ~148k papers published between 2018 and February 2024 that cite one or more of the curated LLMs. This dataset includes structured full texts (when available), titles, abstracts, author information, citation contexts, and metadata such as year, venue, and field of study. In addition, we extracted ~273k citation instances from these papers to the curated LLM set.

### Field classification

To situate LLM adoption across disciplines, we relied on Semantic Scholar's S2FOS3 field-of-study classifier, which assigns each paper to one or more of the 22 fields with a reported accuracy of 86%. The 22 fields outside CS span natural sciences, social sciences, and the humanities. When a paper was associated with multiple fields, it contributed proportionally to each.

For author-level analyses, we identified each researcher's primary field as the domain in which they most frequently published. This allowed us to distinguish collaboration patterns between CS and non-CS researchers. While misclassifications are possible, aggregate analyses across large datasets mitigate their impact, and robustness checks using author primary fields yielded consistent trends.

## Analytical procedures

Our analyses combined quantitative citation statistics with content analysis to address each research question.

(A) **Breadth of LLM Adoption:** We quantified the extent of LLM citations across non-CS fields by calculating the share of citations per field relative to the total. To measure inequality in distribution, we computed the Gini index, where values closer to 0 indicate more even diffusion. We further normalized citation counts by overall publication volume in each field to assess depth of adoption. For historical context, we compared LLM uptake with prior CS exports (Hidden Markov Models [53], Recurrent Neural Networks [54], and Long Short Term Memories [55]) to contextualize their influence.

(B) **Evolving Usage Patterns:** To capture temporal trends, we analyzed year-wise adoption rates of different LLMs across fields. We measured popularity using the annual average number of citations per model and computed the Mean Age of Citation (mAoC) to estimate how quickly each field engages with newer LLMs. We also examined patterns of interdisciplinary collaboration by calculating the proportion of CS authors collaborating with non-CS researchers in LLM-citing papers. We note that this affiliation-based proxy provides only a first-order approximation of interdisciplinarity, as some non-CS authors may hold CS training or dual appointments.

(C) **Applications of LLMs in non-CS Fields:** To explore contexts of use, we extracted citation contexts, titles, and abstracts mentioning LLMs. We processed these using regex-based heuristics to identify trigrams indicative of tasks (e.g., "*protein function prediction*", "*legal judgment prediction*"). Irrelevant phrases were manually filtered. Additionally, we conducted a keyword search to estimate how frequently ethical risks (e.g., bias, hallucination, reproducibility) are explicitly discussed in these papers, complemented by a small human-annotated subset for validation.

## Heuristic validation

To improve reliability, we conducted manual evaluations. For task extraction, a check of randomly sampled 100 representative trigrams confirmed that over 80% corresponded to meaningful research tasks. To evaluate the reliability of our ethical-risk detection method, we manually examined 100 randomly selected papers addressing ethical concerns related to LLMs. This analysis showed high recall, as only 7 relevant papers discussed such concerns without explicitly mentioning LLMs. We further validated precision by examining 200 randomly selected papers and found no cases where ethical issues were mentioned in the same context as LLMs without being identified by our method. While these evaluations are approximate, they suggest that the heuristics are sufficiently reliable for large-scale descriptive analysis.

## Ethics statement

We clarify that the number of citations used in our analysis should not be employed to diminish any specific field or its investments based on low citation counts. Decisions in science and research should rely on a multifaceted evaluation that considers aspects like popularity, relevance, resources, impact, and geographical and temporal factors. This approach prevents oversimplified interpretations and recognizes the diversity and complexity of research fields and their contributions.

## Data

We initiate our study by compiling a list of 106 LLMs from the Stanford Ecosystem Graphs [51]. This curated compilation includes the names of the LLMs selected based on their number of citations (for details, refer to S1 Appendix). We manually identified and included in our list the papers that introduced these LLMs.

Following [56], we use the Semantic Scholar (S2) dataset [52] to obtain the target papers. Specifically, we include around 148*k* papers from non-CS fields from S2 that cite the above-mentioned LLMs. Thus, our dataset contains structured full texts extracted from the PDFs of these papers along with their metadata, encompassing author details, publication year, publication venue, and field of study. In Table 1, we illustrate our dataset statistics.

We prefer S2 over other sources like the ArXiv dataset due to its broader coverage, including ArXiv papers. Additionally, S2 employs a field-of-study classifier (S2FOS3) based on abstracts and titles to identify the field of study (with 86% accuracy). This dataset also contains information on paper citations, including the time and venue of the first publication of the citation, citation context, and citation intents (S2 dataset overview and statistics are described in S1 Appendix and Table 2).

Further, by analyzing author names, unique author ids, and their associations with published papers in S2, we identify the field where an author has published most frequently, which we consider as *their primary field of interest*.

## Main analyses

We use the dataset described above to examine the degree of LLM adoption in fields outside of NLP and CS, addressing three primary aspects: *the scope of LLM influence* (section Breadth of LLM adoption), *their evolving utilization*(section Evolving usage patterns of LLMs), and finally, *their applications in these fields*(section Applications of LLMs in non-CS fields). Subsequently, we offer a detailed examination of each facet.

## Breadth of LLM adoption

To gauge the scope of LLM influence, we address three research questions: first, we investigate *the extent of LLM adoption in fields beyond NLP and CS*; second, we examine *the depth of LLM adoption within these fields* and finally, we compare *LLM popularity to similar past technologies in non-CS fields*.

(Q1) **How widely have LLMs permeated broader academic disciplines (beyond NLP and CS)?**

**Method.** We examine the number of citations from non-CS papers to our curated set of 106 LLMs. If a citing paper is labeled to be in multiple fields, then it contributes to the citation count in each field. We calculate the percentage of

**Table 1**. Statistics of LLMs and their citing papers.

| LLM Timespan | 2018 – Feb. 2024 |
|---|---|
| # LLMs | 106 |
| # papers citing LLMs | 148,501 |
| # citations to LLMs | 273,030 |

**Table 2**. Overall semantic scholar dataset statistics.

| Timespan | 1965 – Feb. 2024 |
|---|---|
| # paper metadata | 200M |
| # paper abstract | 100M |
| # author metadata | 75M |
| # citation instances | 2.5B |
| # papers full-parsed | 5M |

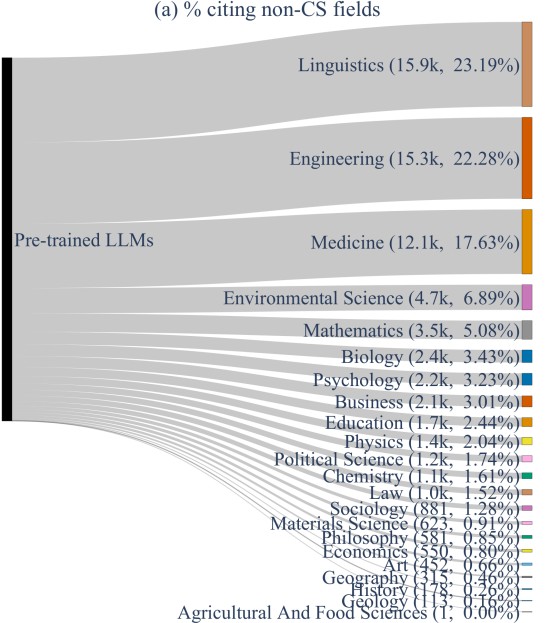

LLM citations attributed to each non-CS field relative to the total citations from all non-CS fields. Further, we use the Gini Index [57] on the relative citation counts to quantify the degree of deviation from a perfectly equal citation distribution across fields. A Gini index of 0 indicates a perfectly uniform distribution, while an index of 1 denotes all the probability mass concentrated on one value.

**Results.** Fig 1(a) is a Sankey plot of the incoming citations from non-CS fields to LLM papers (*#citations*, *%citations*), with the width of the grey flow path representing the citation volume. Certain fields stand out in terms of LLM citations. Linguistics accounts for the maximum number of citations (23.19%), followed by Engineering (22.28%), Medicine (17.63%), Environmental Science (6.89%), and Mathematics (5.08%). Fig 2 reports the Gini index of citation distribution across fields, which has declined over time from 0.77 in 2018 to 0.65 in 2023, suggesting broader adoption.

**Discussion.** Initially, citations were concentrated in Linguistics and Engineering, but over time, LLM usage has spread to more fields. The declining Gini index reflects this diversification. Interestingly, the share of LLM citations by field correlates only weakly with CS collaboration (Pearson 0.28, Fig 1(b)), suggesting that adoption is driven less by interdisciplinary co-authorship and more by perceived utility within fields. To further illustrate the evolving citation trends, in S3 Appendix, we present a year-wise analysis of citations to LLM research papers from non-CS fields.

Further, to assess the depth of LLM penetration across each of these disciplines, examining the variations in publication volumes among these fields is important. In other words, a field with a high number of LLM citations may not be highly influenced by them when viewed in light of the total number of papers published in that field, and vice versa. This guides our exploration of the following section Q2.

**(Q2) To what degree have LLMs penetrated various non-CS fields?**

**Method.** Expanding the analysis in Q1, we now investigate the popularity of LLMs within these fields. For each non-CS field, we calculate the percentage of papers citing LLMs out of the total number of papers published in that field. This acts as a rough indicator of how extensively the field is utilizing LLMs.

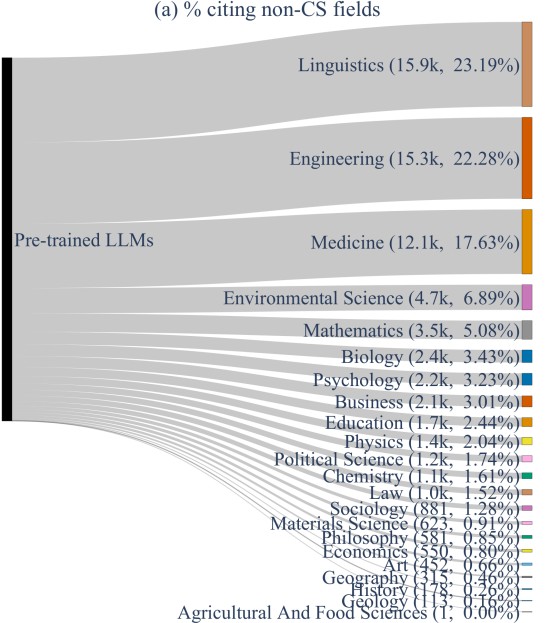
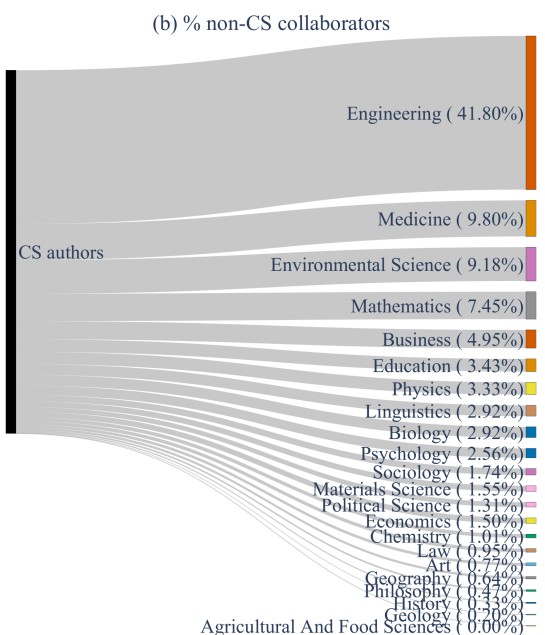

**Fig 1.** **Sankey diagram of LLM citations from non-CS fields.** (a) % citations from non-CS fields to LLMs (left); (b) % CS authors collaborating with non-CS fields (right).

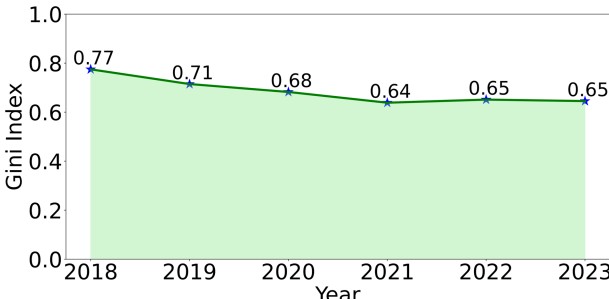

**Fig 2**. **Diversity (Gini index).** Inequality in citation distribution across fields.

**Results.** Fig 3 shows the LLM citation counts relative to the field's publication volume. Proportionally, Linguistics has the most extensive engagement, followed by Law, Mathematics, Art, and Engineering with LLMs compared to other fields. Although Medicine contributes many absolute citations, its proportion of citing papers is smaller compared to fields such as Mathematics and Environmental Science.

**Discussion.** Fields with strong ties to computational methods, such as Mathematics, display relatively high proportional adoption. This reflects both the central role of LLMs as computational models and the tendency of these fields to analyze their underlying properties. By contrast, citation prevalence in Medicine is tempered by the large size of the field, illustrating that absolute citation counts alone may exaggerate impact.

We further analyze the influence of CS author collaborations on research papers within each field by calculating the average number of CS authors per paper within that field since 2018. As illustrated in Fig 4, the fields with the highest average number of CS author influence are Linguistics, Engineering, and Mathematics. We observe a positive Pearson correlation of 0.38 between the average number of CS authors per paper in a non-CS field and the percentage of papers within that field that cite LLMs (indicating moderate correlation). This empirical evidence indicates that although

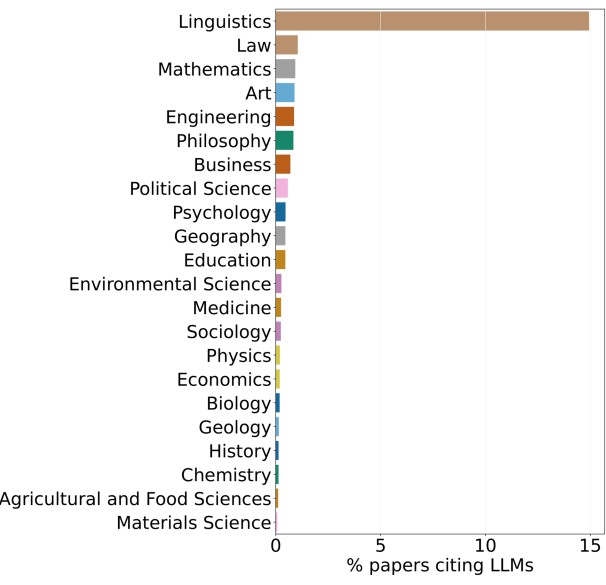

**Fig 3**. **Papers in non-CS fields (Y-Axis) citing LLMs.**

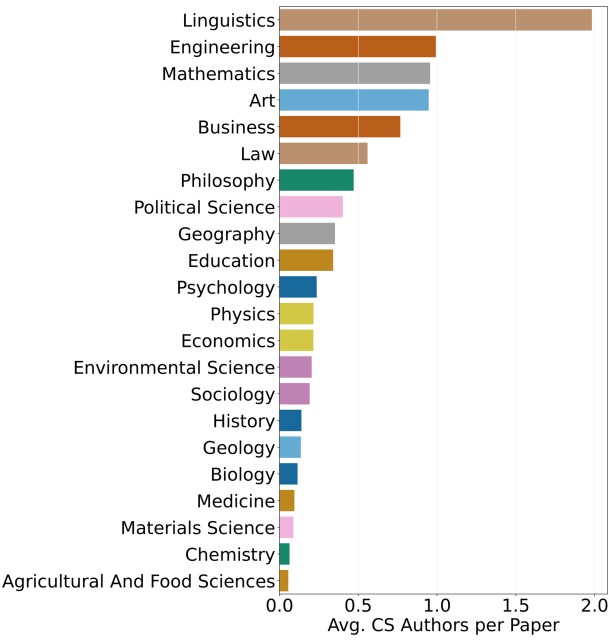

**Fig 4**. **Avg. number of CS authors per paper by field (Y-Axis).**

the total number of CS authors collaborating within a field has little consequence, an increased number of CS authors on a research paper raises the likelihood of using LLMs in that work.

**(Q3) To what extent have non-CS fields embraced LLMs compared to other technologies from CS?**

**Method.** We can gain a better understanding of how markedly LLMs are influencing other fields by comparing their influence to other past CS exports. To explore this, we examine the impact of two iconic technologies: Sequence Models (Hidden Markov Models, Recurrent Neural Networks, and Long Short Term Memories) and Hidden Markov Models (HMMs), whose revolutionary effects in CS subfields mirror the transformative influence of LLMs today and juxtapose their wide impact with that of LLMs. We compile all representative papers for RNNs-LSTMs and HMMs from [58], highly cited not just within NLP but also across other CS sub-fields, and count the citations they receive from non-CS fields using a methodology the same as Q1. To assess the diversity of their adoption, we use Gini indices on the citation distributions for these technologies. A Gini index of 0 indicates a perfectly equal distribution, whereas a value of 1 indicates total concentration in a single technology.

**Results.** In Table 3, we present the diversity indices: HMMs (0.66), RNN/LSTMs (0.63), and LLMs (0.56). LLMs thus show the most even spread of adoption across fields.

**Table 3**. **Diversity (Gini index) in citation distribution across fields for NLP Technologies.**

| Cited Method | Gini Index |
|---|---|
| HMMs | 0.66 |
| RNN-LSTMs | 0.63 |
| LLMs | **0.56** |

**Discussion.** LLMs demonstrate the lowest Gini index among these technologies, suggesting broader adoption across various non-CS fields than earlier CS exports. Further, to compare the popularity of LLMs with previous technologies, we set four different thresholds and show the number of fields surpassing each threshold. Fig 5 shows that LLMs receive citations more broadly across different fields, emphasizing their wider range of applications (refer to S3 Appendix) for additional analyses.

## Evolving usage patterns of LLMs

To study how LLM utilization across various fields has evolved, we address two research questions: first, we identify *the most widely used LLMs in these fields*; second, we analyze *which fields have embraced newer LLMs over time*.

**(Q4) What are the most popular LLMs in different non-CS fields, and why?**

**Method.** We evaluate the popularity of LLMs across non-CS fields by computing the annual average number of citations each LLM receives from these fields.

**Results.** Fig 6 shows the top 25 most popular LLMs across diverse non-CS fields (plot for a larger list of LLMs in S3 Appendix). Below, we summarize the main findings.

**Discussion.** BERT, despite being officially published in 2019 (relatively early in the evolution of LLMs), remains the most popular LLM among non-CS fields, with the highest average citations per year. This could be because when BERT was introduced, there were fewer LLM options, leading to more citations. Consequently, many papers that incorporated LLMs during that period likely cited BERT, contributing to its higher average citation count. Secondly, the extensive, wide, and successful use of BERT likely made it a reliable choice for non-CS fields. In contrast, the behaviors and capabilities of newer LLMs are still undergoing exploration and evaluation.

In Fig 7, we show the number of LLMs published in each year. We observe that, in recent years, the proliferation of LLM research has led to the emergence of numerous new LLMs, and by February 2024, we already see citations beginning to shift from earlier dominant models toward these then-recent entrants (e.g., GPT-3, GPT-2, LLaMA, T5).

**(Q5) What is the average LLM citation age in non-CS fields, and how does it differ across fields?**

**Method.** The LLM citation age represents the average age of the LLMs that papers in a particular field are citing. A low citation age indicates a preference for newer LLMs, while a high age suggests reliance on older ones. Fields falling in between use a mix of both.

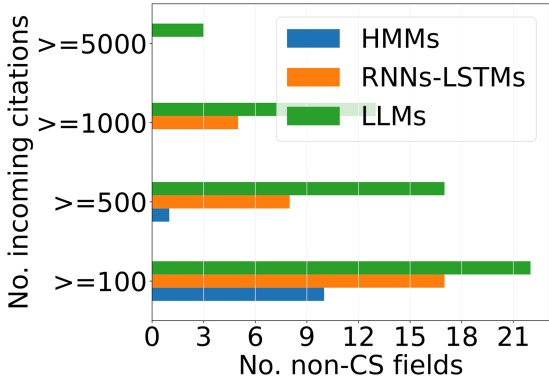

**Fig 5. Citations across NLP technologies.** Number of fields surpassing the threshold number of citations for different NLP Technologies.

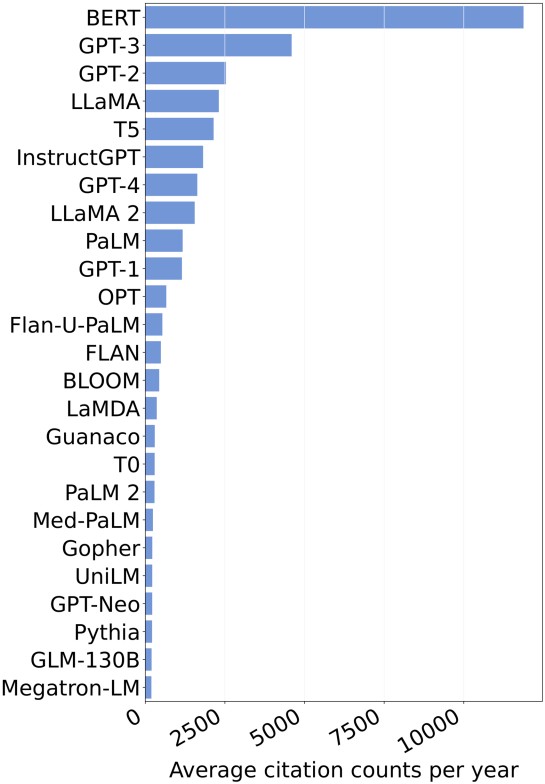

**Fig 6**. Popular LLMs in non-CS fields (Top 25).

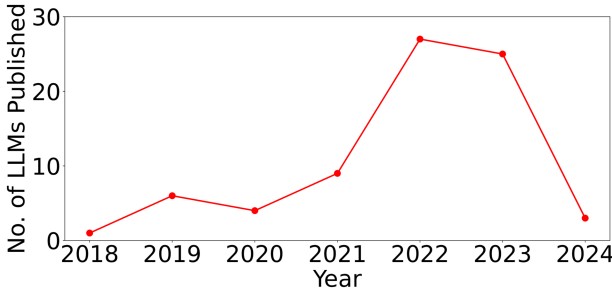

**Fig 7**. No. of LLMs Published over the Years (until Feb.'24).

If a paper $x_i$ from research field $c$ (i.e., $x_i \in c$) cites an LLM paper $y_j$, then the age of LLM citation (AoC) is taken to be the difference between the year of publication (YoP) of $x_i$ and $y_j$:

$$AoC(x_i, y_j) = YoP(x_i) - YoP(y_j) \qquad (1)$$

We calculate the mean Age of LLM Citation ($mAoC_c$) for a field $c$ as:

$$mAoC_c = \frac{1}{MN} \sum_{i=1}^{M} \sum_{j=1}^{N} AoC(x_i, y_j), \forall x_i \in c \qquad (2)$$

**Results.** Fig 8 shows that Psychology and Linguistics cite relatively recent LLMs (low mAoC), while Environmental Science relies more on older models. Biology and Chemistry strike a balance between older and newer models.

**Discussion.** These findings suggest that some fields (e.g., Psychology) rapidly integrate newer architectures, while others (e.g., Environmental Science) continue to depend on established models.

Psychology and Linguistics, having a longer history with LLMs, are now exploring newer LLMs, leading to low mAoC. Environmental Science relies mainly on older LLMs, possibly due to established performance and a lack of LLMs specifically created for the field. However, fields like Biology and Chemistry keep a balance between older LLMs while embracing newer LLMs, which are suited for their task, leading to moderate mAoC.

## Applications of LLMs in non-CS fields

To examine LLM applications in non-CS fields, *we qualitatively identify tasks where LLMs are applied*, and additionally, *using keyword searches, we roughly gauge the frequency with which the papers that cite LLMs in these fields mention ethical risks*.

### (Q6)  Do non-CS fields primarily fine-tune LLMs or use them in inference/zero-shot setting?

**Method.** To estimate how LLMs are used in practice, we searched citing papers for mentions of *fine-tune*, *zero-shot*, and *inference* within abstracts. This filtering yielded 3,675 papers mentioning one of these terms. We then conducted manual validation to estimate the reliability of these signals. Specifically, we randomly sampled 50 papers mentioning fine-tuning and 100 papers mentioning inference/zero-shot to assess whether the terms were used in the context of actually applying LLMs.

**Results.** Among the 3,675 papers mentioning either fine-tuning or inference, 479 explicitly referenced fine-tuning and 3,196 mentioned inference or zero-shot usage. Manual validation showed that 43 of 50 sampled fine-tuning papers (precision ≈ 86%) indeed described adapting LLMs for domain-specific tasks, while 80 of 100 sampled inference papers (precision ≈ 80%) genuinely reported applying LLMs without retraining. Overall, papers describing inference usage outnumbered those describing fine-tuning by a factor of approximately 6.6 to 1, indicating that inference-based adoption is by far the dominant practice in non-CS fields.

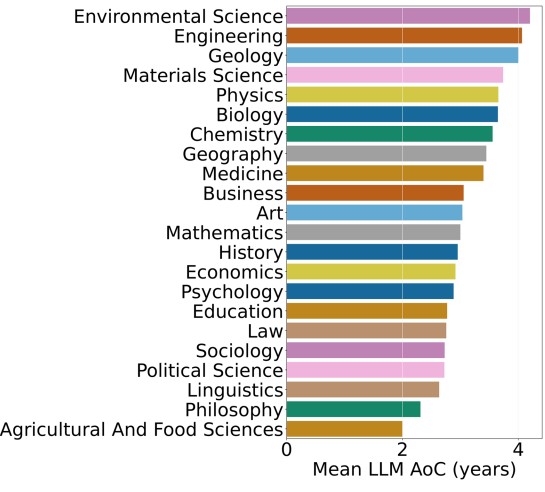

**Fig 8**. **Age-of-Citation (AoC) of non-CS fields (Y-Axis).**

**Discussion.** These findings suggest that non-CS fields overwhelmingly adopt LLMs as task-agnostic tools in inference or zero/few-shot settings rather than investing resources in fine-tuning. This aligns with the preference for models like GPT-3 or LLaMA, which can be readily applied across tasks without domain-specific retraining. The slightly lower precision in detecting inference cases reflects the broader and more varied use of the term "inference" across disciplines, but the manual validation confirms that the dominant pattern holds robustly at scale. Taken together, this analysis complements our earlier observation (Q5) that non-CS fields favor adaptable, off-the-shelf models rather than fine-tuned variants.

## (Q7) How are LLMs utilized in non-CS fields?

**Method.** Understanding these field-specific tasks, where LLMs find applications, can help the NLP community better understand, adapt, and align LLMs and research with the distinct requirements of these communities. We examine LLM usage in various non-CS fields using trigrams in paper titles, abstracts, and citation contexts. By citation contexts, we precisely mean sentences that explicitly mention LLMs by name or citation [59]. We further process them to identify trigrams indicative of tasks using regex-based heuristics, such as removing the stop-words and manually filtering out irrelevant trigrams. Unlike [17], we use trigrams over bigrams, as trigrams offer richer contextual information beneficial for task identification within each field.

**Results.** In Table 4, we highlight the top three most frequent trigrams that serve as representative tasks in papers citing LLMs for each of the non-CS fields. This table offers a glimpse into the specific problems LLMs are employed to address across various fields. In Table 5, we present frequent bigrams from these research papers, offering insights into the topics that interest these fields.

**Discussion.** Non-CS fields employ LLMs primarily to solve domain problems rather than to analyze the models themselves, suggesting that LLMs are being integrated as tools into disciplinary workflows (Refer to S3 Appendix for additional analysis).

**Table 4**. Most frequent task trigrams in LLM citing papers from non-CS fields.

| field-of-study | Frequent Task representative Trigrams | | |
|---|---|---|---|
| *Biology* | protein function prediction | protein structure prediction | protein sequence design |
| *Chemistry* | compound-protein interaction prediction | molecular properties prediction | molecular dynamics simulations |
| *Psychology* | mental health support | detecting rating humor | humor offense rating |
| *Environmental Science* | hyperspectral image classification | image change detection | land cover classification |
| *Law* | legal judgment prediction | legal case retrieval | legal case matching |
| *Art* | visual question answering | content style text-to-drawing | design concept generation |
| *Sociology* | whose heritage classification | daily conversational analysis | discourse relation recognition |
| *Business* | stock price prediction | named entity recognition | graphic layout generation |
| *Philosophy* | event causality classification | explainable causal reasoning | cheat turing test |
| *Linguistics* | grapheme to phoneme | language culture internalization | zero-shot cross-lingual transfer |
| *Mathematics* | sensing scene classification | land cover classification | stochastic differential equations |
| *Physics* | low-light image enhancement | quantum machine learning | multi-speaker speech synthesis |
| *Education* | educational question generation | automated essay scoring | generation reading comprehension |
| *Economics* | assistive response generation | financial statement analysis | stock price prediction |
| *Geology* | seismic data interpolation | seismic phase picking | simulation seismic waves |
| *Engineering* | short-term load forecasting | non-intrusive load monitoring | energy consumption forecasting |
| *Medicine* | radiology report generation | optical coherence tomography | clinical decision support |
| *Geography* | geographic language understanding | spatially-explicit machine learning | geo-spatial knowledge graphs |
| *Political Science* | monitoring public discussion | processing social psychology | reframed multilingual analysis |
| *Agriculture And Food Science* | foaming structural studies | rice yield prediction | estimating grape yield |
| *Materials Science* | material property prediction | crystal structure generation | functional materials discovery |
| *History* | historical event extraction | ancient latin inscription | reconstruct ancient mosaics |

**Table 5.** Most frequent bigrams identifying dominant areas in LLM citing papers from non-CS fields.

| field-of-study | Frequent bigrams representative of areas of interest | | |
|---|---|---|---|
| Biology | protein structure | structure prediction | structure database |
| Chemistry | structural basis | molecular dynamics | structural insights |
| Psychology | social media | mental health | emotion recognition |
| Environmental Science | sensing images | remote sensing | object detection |
| Law | judgment prediction | legal case | legal reasoning |
| Art | stable diffusion | text-to-image generation | text-to-image synthesis |
| Sociology | case study | cultural heritage | political ideology |
| Business | stock price | social media | layout generation |
| Philosophy | moral code | event causality | causal reasoning |
| Linguistics | grapheme phoneme | cross lingual | social bias |
| Mathematics | hyperspectral image | inverse problems | differential equations |
| Physics | low-light image | quantum neural | energy physics |
| Education | question generation | keyword extraction | essay scoring |
| Economics | policy uncertainty | economic policy | price prediction |
| Geology | seismic phase | seismic waves | phase picking |
| Engineering | load forecasting | load monitoring | fault diagnosis |
| Medicine | health records | radiology reports | clinical notes |
| Geography | geographic language | transient chaos | geospatial knowledge |
| Political Science | news media | public discussion | news articles |
| Agriculture And Food Science | gene family | yield prediction | drought tolerance |
| Materials Science | property prediction | crystal structure | feature fusion |
| History | historical event | latin inscription | ancient mosaics |

## (Q8) How frequently do research papers in non-CS fields mention ethical risks?

**Method.** Until the preceding sections, we studied the impact of LLMs on non-CS fields and their utilization trends. However, it is essential to acknowledge that LLMs come with inherent risks, such as bias and hallucination [60]. These concerns are particularly relevant when LLMs are applied to sensitive applications.

While we acknowledge that pinpointing research papers that address ethical concerns about LLMs from a large collection presents a significant challenge (and requires huge manual efforts), we argue that identifying the contexts in which these papers discuss risks and ethical concerns can provide a rough idea and preliminary insights into how deeply non-CS fields are engaged with the ethical implications of LLMs. Hence, we examine papers that cite LLMs, aiming to identify contexts where discussions of risks and ethical considerations closely coincide with mentions of LLMs within the same sentence. Additionally, we extend our scrutiny to the titles and abstracts of these papers, assessing whether these concerns are underscored as particularly significant. Specifically, we searched for keywords related to ethical risks (e.g., "bias," "hallucination," "ethics") and validated the method through manual checks of 100 flagged and 200 unflagged papers.

**Results.** In Fig 9, we present the percentages of papers in each field that mention the risks associated with LLMs in their titles, abstracts, or citation contexts at least once relative to the total number of papers citing LLMs in that field. This acts as an *approximate indicator* of the relative importance of ethical considerations within various non-CS fields. To assess the reliability of our ethical-risk detection method, we manually reviewed 100 randomly selected papers on ethical concerns related to LLMs. The analysis indicated high recall, with only 7 relevant papers raising such concerns without explicitly referencing LLMs, confirming reliability. Refer to S2 Appendix for a detailed description and manual evaluation of this analysis method.

Our analysis reveals that, on average, approximately 2.01% of non-CS papers citing LLMs mention the ethical risks of LLMs. This finding is concerning, as it suggests that many non-CS fields are actively using LLMs without recognizing the ethical risks that these models entail.

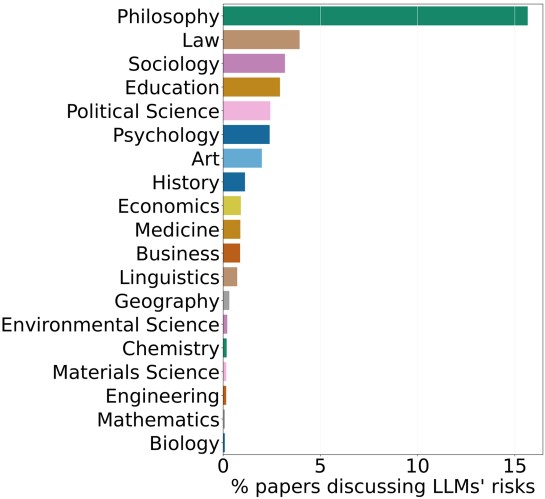

**Fig 9**. **LLM risk discussions by field.** % of papers per field (Y-Axis) discussing LLM risks.

**Discussion.** Despite growing adoption, explicit acknowledgment of ethical risks remains rare outside a few disciplines, pointing to a substantial awareness gap as LLMs are applied in sensitive domains such as Medicine and Psychology.

## Discussion: Opportunities and challenges

Our study provides the first large-scale descriptive baseline of LLM diffusion across disciplines beyond Computer Science. Through this study, we explored the broad diffusion of LLMs across academic disciplines beyond CS, highlighting their influence on fields as varied as Linguistics, Engineering, Medicine, and Law. By analyzing citation data and usage patterns from over 148k papers across 22 non-CS domains, we learned several important lessons about the evolving nature of LLM usage across scholarly research.

Firstly, we observe that LLMs have gained significant traction across multiple non-CS fields, with Linguistics, Engineering, and Medicine being the primary domains of citation. The growing number of LLM citations over time reflects the increasing adoption of these models, particularly those capable of zero-shot and few-shot learning, which do not require fine-tuning for domain-specific tasks. This widespread adoption highlights the versatility of LLMs in addressing diverse challenges, from protein function prediction in Biology to legal case retrieval in Law.

However, despite the broad use of LLMs, our study uncovers disparities in how various fields engage with this technology. Fields that are traditionally closer to NLP, like Linguistics, tend to exhibit higher levels of adoption, whereas other fields, such as Sociology or Philosophy, show more cautious or less frequent usage. This variation points to the need for more tailored LLM models that can better meet the specific needs of these fields.

Moreover, while LLMs are becoming increasingly popular, they come with their own set of concerns, particularly ethical issues such as the potential for biased or misleading outputs (and even hallucinations). Our analysis reveals that discussions around these risks are notably absent in many non-CS papers that cite LLMs, which raises alarms about the lack of awareness (consequently, mitigation strategies) across various disciplines. Fields like Philosophy and Law are exceptions, where ethical considerations are more actively addressed, but the overall trend indicates an urgent need for better integration of ethical frameworks in LLM usage.

As we look toward the future, it is clear that LLMs will continue to shape interdisciplinary research. The increasing volume of LLM citations across disciplines suggests that these models will not only augment existing research methods but also facilitate entirely new lines of inquiry. However, for this potential to be fully realized, it is crucial that the broader

academic community, especially those in non-CS fields, become more educated about both the capabilities and risks associated with LLMs. Furthermore, there is a pressing need for the development of domain-specific LLMs that are better suited to the unique challenges of fields such as Medicine, Psychology, and Environmental Science.

In conclusion, while the integration of LLMs into non-CS research is undeniably a promising development, it brings with it both exciting opportunities and significant challenges. Researchers and practitioners must engage in responsible AI practices, fostering interdisciplinary collaboration and a deeper understanding of the ethical implications of LLMs. Recommendations include further investigation into the ethical aspects of LLM usage, the creation of tailored models for specific disciplines, and promoting awareness about the limitations of these models in handling complex, domain-specific tasks. By addressing these concerns, we can ensure that LLMs contribute to the advancement of knowledge in a safe, effective, and ethical manner.

### Key messages

Our findings highlight several important takeaways for understanding the diffusion of LLMs across non-CS fields:

1. **Broad diffusion with disciplinary variation.** LLMs have diffused widely beyond Computer Science, with Linguistics, Engineering, and Medicine emerging as the most active adopters. Fields more closely aligned with CS tend to adopt faster, whereas others show slower or more selective uptake.
2. **Enduring influence of early models.** BERT remains the most enduringly cited LLM, while task-agnostic models such as GPT-3 and LLaMA are increasingly favored for domain-specific applications due to their zero/few-shot capabilities.
3. **Predominantly applied to domain tasks.** Non-CS fields typically use LLMs to solve their own domain problems—such as protein function prediction, legal judgment retrieval, or radiology report generation—rather than analyzing the models themselves.
4. **Gaps in ethical awareness.** Only a small fraction of papers outside CS explicitly acknowledge ethical concerns such as bias, hallucination, or reproducibility. This highlights a pressing need for broader awareness and integration of responsible AI practices across disciplines.
5. **Opportunities for tailored models.** The variation in adoption patterns underscores the potential of developing domain-adapted or instruction-tuned LLMs better suited to the needs of Medicine, Environmental Science, Psychology, and other fields.

### Threats to validity

While our study provides systematic evidence of the diffusion of LLMs across academic fields, several threats to validity should be considered when interpreting the findings:

- **Citation data biases.** Citations serve as a rough proxy for influence but do not always indicate direct methodological adoption. However, citation practices also differ across disciplines, with some fields citing more heavily than others. Although we partly mitigate this by reporting field-normalized proportions (e.g., share of papers citing LLMs within a field), these metrics cannot fully control for disciplinary citation cultures.
- **Model selection bias.** Our curated set of 106 LLMs prioritizes widely cited foundation models with identifiable publications. More recent instruction-tuned, multi-modal, or domain-specific LLMs are underrepresented because of limited citation coverage and uneven indexing in bibliographic databases.
- **Field classification accuracy.** Field labels were assigned using the S2FOS3 classifier, which has a reported accuracy of ~86%. Misclassifications are inevitable and may affect fine-grained field statistics. To reduce noise, our analyses emphasize aggregate trends, and robustness checks with author primary fields yielded consistent patterns. Nonetheless, results should be interpreted at a field level rather than as precise counts for individual papers.

- **Global and linguistic disparities.** Semantic Scholar primarily indexes English-language and Western venues, which likely under-represents adoption in non-English research communities and regions with limited coverage.
- **Measuring interdisciplinarity.** Our proxy for interdisciplinarity—co-authorship with CS-affiliated authors—offers only a first-order approximation. Some non-CS researchers may hold CS training or dual appointments, which could inflate estimates of collaboration. More robust alternatives, such as cross-field publication history or venue diversity, remain promising directions for future work.
- **Heuristics methods.** Tasks extraction and ethical risk detection rely on heuristics and keyword searches. While small-scale manual validations suggest high precision and recall, these methods remain approximations. Noise in automatic extraction is expected, but aggregate-level results are robust enough to indicate broad trends.

By acknowledging these threats, we aim to contextualize our results as robust at the aggregate level but not immune to the biases inherent in large-scale scientometric analyses.

## Supporting information

**S1 Appendix. Supplementary definitions and discussions.**
(PDF)

**S2 Appendix. Human evaluation of LLM citing papers that discuss LLM risks.**
(PDF)

**S3 Appendix. Additional results.**
(PDF)

## Author contributions

**Conceptualization:** Aniket Pramanick, Yufang Hou, Saif M. Mohammad.

**Data curation:** Aniket Pramanick.

**Formal analysis:** Aniket Pramanick.

**Funding acquisition:** Iryna Gurevych.

**Investigation:** Aniket Pramanick, Yufang Hou.

**Methodology:** Aniket Pramanick, Saif M. Mohammad, Yufang Hou.

**Project administration:** Yufang Hou, Saif M. Mohammad, Iryna Gurevych.

**Resources:** Iryna Gurevych.

**Software:** Aniket Pramanick.

**Supervision:** Yufang Hou, Saif M. Mohammad, Iryna Gurevych.

**Validation:** Saif M. Mohammad.

**Writing – original draft:** Aniket Pramanick.

**Writing – review & editing:** Yufang Hou, Saif M. Mohammad, Iryna Gurevych.

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
