## [Decision Letter · Decision Letter 0]

31 Jul 2025

PONE-D-25-35979

Transforming Scholarly Landscapes: The Influence of Large Language Models on Academic Fields beyond Computer Science

PLOS ONE

Dear Dr. Pramanick,

Thank you for submitting your manuscript to PLOS ONE. After careful consideration, we feel that it has merit but does not fully meet PLOS ONE’s publication criteria as it currently stands. Therefore, we invite you to submit a revised version of the manuscript that addresses the points raised during the review process.

Please adequately address all the concerns and questions raised by the reviewers

We look forward to receiving your revised manuscript.

Kind regards,

Bekalu Tadesse Moges

Academic Editor

PLOS ONE

Journal Requirements:

This work has been funded by the German

Research Foundation (DFG) as part of the Research Training Group KRITIS No. GRK 2222.

4. Please expand the acronym “DFG” (as indicated in your financial disclosure) so that it states the name of your funders in full.

Reviewers' comments:

Reviewer's Responses to Questions

**Comments to the Author**

1. Is the manuscript technically sound, and do the data support the conclusions?

Reviewer #1: Yes

Reviewer #2: Partly

2. Has the statistical analysis been performed appropriately and rigorously?

Reviewer #1: Yes

Reviewer #2: No

3. Have the authors made all data underlying the findings in their manuscript fully available?

Reviewer #1: Yes

Reviewer #2: Yes

4. Is the manuscript presented in an intelligible fashion and written in standard English?

Reviewer #1: Yes

Reviewer #2: Yes

5. Review Comments to the Author

Reviewer #1: The paper presents an interesting and novel approach to tracing the diffusion and application of LLMs—particularly traditional ones like BERT—across 22 academic fields outside the domain of CS. It effectively addresses two core research questions: (1) the breadth of LLM adoption in non-CS disciplines, and (2) their evolving temporal usage patterns. The analysis also includes a qualitative investigation of usage adaptations, enriching the contribution.

I appreciate the paper’s clear and straightforward writing style, which makes it accessible to a broad audience. However, this clarity comes at the cost of under specification in some methodological and scoping decisions. For example:

• Why were only traditional LLMs such as BERT included, while more recent architectures (e.g., instruction-tuned or multimodal models) are excluded, especially considering their growing adoption? Is the selection of LLMs skewed toward older or better-documented models? Are recent instruction-tuned or domain-specific LLMs underrepresented?

• Why were only 22 fields selected, and what criteria guided this selection? While these fields do reflect diverse applications, the limitation needs further justification.

• Is citation count a sufficient and reliable proxy for measuring actual influence or usage of LLMs within a field? How are citation intents (e.g., methodological adoption vs. general reference) distinguished?

• Are all citations given equal weight regardless of field norms or citation behaviors? For instance, fields like medicine or law might have very different citation cultures compared to computer science or linguistics.

• How accurate is the automatic classification of a paper into a given field? Were any steps taken to correct or validate misclassified papers?

• Does the paper account for global or linguistic disparities in LLM adoption across fields?

• The paper uses co-authorship with CS-affiliated individuals as a proxy for interdisciplinary collaboration. However, this assumption is problematic and potentially misleading. Many researchers in fields like engineering or linguistics may hold CS degrees, have dual appointments, or regularly publish in CS venues despite being officially affiliated with non-CS departments. Therefore, co-authorship with a "CS author" does not necessarily indicate interdisciplinary collaboration in practice—it may simply reflect intra-disciplinary work by CS-trained researchers operating in adjacent fields. So, why did the authors rely solely on affiliation-based metrics for measuring interdisciplinarity, and have they considered more robust alternatives? For example, analysis of cross-field publication history, topic modeling, or venue diversity could yield a more accurate picture of true cross-disciplinary exchange. Without this, the conclusions about collaboration patterns may be significantly overstated or skewed.

To improve transparency and rigor, I strongly recommend adding a dedicated section titled “Research Methodology” or “Research Methods” .. This section should explicitly describe how each research question was addressed, including data collection, field classification, and analysis procedures. Additionally, a “Threats to Validity” section would be important for addressing potential biases in citation data, model selection, and field delineation.

The discussion section is thought-provoking and contains valuable insights. However, transforming these insights into a concise “Takeaways” or “Key Messages” subsection would enhance readability and impact for the reader.

I thoroughly enjoyed reading this paper and commend the authors for their efforts. I believe this work has good potential, and with further revisions, it can make a valuable contribution to understanding the interdisciplinary reach of LLMs.

Reviewer #2: Thank you for allowing me to review this manuscript. The manuscript investigates the influence of LLMs through a large-scale citation analysis of approximately 148k non-CS papers citing 106 LLMs. The authors examine citation patterns, usage contexts, and ethical discourse across 22 fields using data from Semantic Scholar. The study addresses a timely and important topic. This manuscript presents a valuable descriptive contribution and provides new insights into the diffusion of LLMs across non-CS fields. However, it requires significant revisions, particularly methodological and statistical reliability. I have multiple concerns about this paper:

1. While the manuscript addresses important questions, its contribution is primarily descriptive. It reads more as a survey of citation trends due to the reliance on standard bibliometric techniques (e.g., citation counts, Gini index). I recommend reframing the manuscript as a descriptive exploratory study or providing empirical depth analysis to reflect the systematic and transformative impact analysis.

2. The methodology is generally suitable for large-scale citation analysis, but it suffers from statistical reliability and reproducibility. Specifically, the analysis is limited to descriptive statistics, with no inferential testing to assess the significance or strength of observed patterns. The task extraction process based on trigrams, as well as the heuristic detection of ethical mentions, are not systematically evaluated in the main text. I recommend incorporating statistical modeling techniques, such as regression or hypothesis testing, validating text-mining heuristics using standard metrics (e.g., precision, recall, F1).

3. The "transformative influence" of LLMs is overstated based on the descriptive nature of the evidence. Citation counts and trigram frequencies do not directly demonstrate transformation, influence, or practical impact in these fields. I recommend rephrasing or qualifying claims about "impact" and "influence" to more accurately reflect the descriptive scope of the study.

4. The manuscript is generally well-written, and the structure is coherent. However, there are inconsistencies in the formatting of results and discussion sections, and some key terms (e.g., "task-agnostic models") are used without definition. I recommend standardizing the Results/Discussion structure across sections and clarifying key concepts upon first use.

5. The manuscript refers in S1 Appendix to a curated list of 106 LLMs derived from the Stanford CRFM Ecosystem Graph. However, it is unclear what criteria were used to include or exclude models. Was there a minimum citation threshold or usage filter? Was the selection random or top-down based on influence?

6. What is the difference between # papers citing LLMs and # citations to LLMs in Table 1? It is confusing.

7. S1 and S2 Appendices are valuable, offering critical context and validation for the study’s data and methods. I suggest considering incorporating key elements of S1 and S2 into the main manuscript.

6. PLOS authors have the option to publish the peer review history of their article (what does this mean?). If published, this will include your full peer review and any attached files.

Reviewer #1: No

Reviewer #2: No

---

## [Author Response · Author response to Decision Letter 1]

4 Oct 2025

The response letter addressing the reviewers’ comments is included alongside the updated manuscript.

---

## [Decision Letter · Decision Letter 1]

5 Nov 2025

Transforming Scholarly Landscapes: The Influence of Large Language Models on Academic Fields beyond Computer Science

PONE-D-25-35979R1

Dear Dr. Pramanick,

We’re pleased to inform you that your manuscript has been judged scientifically suitable for publication and will be formally accepted for publication once it meets all outstanding technical requirements.

Kind regards,

Bekalu Tadesse Moges

Academic Editor

PLOS ONE

Additional Editor Comments (optional):

Reviewers' comments:

Reviewer's Responses to Questions

**Comments to the Author**

1. If the authors have adequately addressed your comments raised in a previous round of review and you feel that this manuscript is now acceptable for publication, you may indicate that here to bypass the “Comments to the Author” section, enter your conflict of interest statement in the “Confidential to Editor” section, and submit your "Accept" recommendation.

Reviewer #1: All comments have been addressed

Reviewer #2: All comments have been addressed

2. Is the manuscript technically sound, and do the data support the conclusions?

Reviewer #1: Yes

Reviewer #2: Yes

3. Has the statistical analysis been performed appropriately and rigorously?

Reviewer #1: Yes

Reviewer #2: Yes

4. Have the authors made all data underlying the findings in their manuscript fully available?

Reviewer #1: Yes

Reviewer #2: Yes

5. Is the manuscript presented in an intelligible fashion and written in standard English?

Reviewer #1: Yes

Reviewer #2: Yes

6. Review Comments to the Author

Reviewer #1: (No Response)

Reviewer #2: The authors have substantially improved the manuscript in response to prior feedback. The revised version clearly reframes the study as a descriptive and systematic mapping of LLM diffusion across non-CS fields. Methodological transparency has been strengthened through added validation metrics for heuristic processes and the inclusion of detailed Research Methodology and Threats to Validity sections. Overall, the revision demonstrates rigor, clarity, and responsiveness. I am satisfied that all my concerns have been addressed, and I recommend acceptance of the manuscript in its current form.

7. PLOS authors have the option to publish the peer review history of their article (what does this mean?). If published, this will include your full peer review and any attached files.

Reviewer #1: No

Reviewer #2: No

---

## [Editor Report · Acceptance letter]

PONE-D-25-35979R1

PLOS ONE

Dear Dr. Pramanick,

I'm pleased to inform you that your manuscript has been deemed suitable for publication in PLOS ONE. Congratulations! Your manuscript is now being handed over to our production team.

Kind regards,

on behalf of

Dr. Bekalu Tadesse Moges

Academic Editor

PLOS ONE